# Vitamin D and Metabolic Dysfunction-Associated Steatotic Liver Disease (MASLD): Novel Mechanistic Insights

**DOI:** 10.3390/ijms25094901

**Published:** 2024-04-30

**Authors:** Ioanna Aggeletopoulou, Efthymios P. Tsounis, Christos Triantos

**Affiliations:** Division of Gastroenterology, Department of Internal Medicine, University Hospital of Patras, 26504 Patras, Greece; iaggel@upatras.gr (I.A.); makotsouno@gmail.com (E.P.T.)

**Keywords:** vitamin D, vitamin D receptor, metabolic dysfunction-associated steatotic liver disease, MASLD, inflammation, insulin resistance, hepatic steatosis

## Abstract

Metabolic dysfunction-associated steatotic liver disease (MASLD) is an increasingly prevalent condition characterized by abnormal fat accumulation in the liver, often associated with metabolic disorders. Emerging evidence suggests a potential link between vitamin D deficiency and the development and progression of MASLD. The current review provides a concise overview of recent studies uncovering novel mechanistic insights into the interplay between vitamin D and MASLD. Several epidemiological studies have highlighted a significant association between low vitamin D levels and an increased risk of MASLD. Vitamin D, traditionally known for its role in bone health, has now been recognized as a key player in various physiological processes, including immune regulation and inflammation. Experimental studies using animal models have demonstrated that vitamin D deficiency exacerbates liver steatosis and inflammation, suggesting a potential protective role against MASLD. Mechanistically, vitamin D appears to modulate MASLD through multiple pathways. Firstly, the vitamin D receptor (VDR) is abundantly expressed in liver cells, indicating a direct regulatory role in hepatic function. Activation of the VDR has been shown to suppress hepatic lipid accumulation and inflammation, providing a mechanistic basis for the observed protective effects. Additionally, vitamin D influences insulin sensitivity, a critical factor in MASLD pathogenesis. Improved insulin sensitivity may mitigate the excessive accumulation of fat in the liver, thus attenuating MASLD progression. In parallel, vitamin D exhibits anti-inflammatory properties by inhibiting pro-inflammatory cytokines implicated in MASLD pathophysiology. Experimental evidence suggests that the immunomodulatory effects of vitamin D extend to the liver, reducing inflammation and oxidative stress, key drivers of MASLD, and the likelihood of hepatocyte injury and fibrosis. Understanding the complex interplay between vitamin D and MASLD provides a basis for exploring targeted therapeutic strategies and preventive interventions. As vitamin D deficiency is a modifiable risk factor, addressing this nutritional concern may prove beneficial in mitigating the burden of MASLD and associated metabolic disorders.

## 1. Introduction

Non-alcoholic fatty liver disease (NAFLD) encompasses a spectrum of chronic liver conditions, ranging from simple steatosis, characterized by at least 5% fat accumulation in liver imaging or histological assessments, to non-alcoholic steatohepatitis (NASH), marked by inflammation associated with steatosis, progressing to fibrosis and established cirrhosis [1]. A serious complication of NAFLD is the potential development of hepatocellular carcinoma (HCC). In 2023, a consensus group comprising multiple medical societies redefined the nomenclature for NAFLD to metabolic dysfunction-associated steatotic liver disease (MASLD) [2,3]. This adjustment aimed to depict the underlying etiological pathways of the condition more accurately and reduce potential stigmatization [2,4]. MASLD is defined as the presence of hepatic steatosis along with at least one cardiometabolic risk factor [5]. MASLD has been estimated to impact 30% of the adult population globally, showing an increase in prevalence from 22% to 37% between 1991 and 2019 [6,7]. This increase aligns with the growing prevalence of obesity and obesity-related diseases. The more severe form of MASLD, known as metabolic dysfunction-associated steatohepatitis (MASH), is histologically defined as the presence of lobular inflammation and hepatocyte ballooning and is linked to a higher risk of fibrosis progression [8,9,10]. Cardiovascular disease (CVD) stands as the primary cause of mortality in MASLD patients. However, those with more advanced liver fibrosis deal with an elevated risk of liver-related mortality, which rises exponentially in correlation with the progression of the fibrosis stage [11]. A prospective study on a MASLD patient cohort with baseline liver biopsy revealed a cardiovascular event rate of 2.03 per 100 person-years and a liver-related event rate of 0.43 per 100 person-years [12].

The potential effects of vitamin D on impeding the progression of chronic liver disease and, more specifically, that of MASLD are broadly discussed due to the emerging antiproliferative, anti-inflammatory, and anti-fibrotic properties of this nutrient [13,14,15,16]. The aim of the current review is to comprehensively examine and elucidate the relationship between vitamin D deficiency and the vitamin D-vitamin D receptor (VDR) axis in the context of MASLD pathogenesis and progression. This includes investigating the molecular mechanisms through which vitamin D influences key aspects of MASLD, such as hepatic lipid metabolism, inflammation, insulin sensitivity, and fibrosis. Additionally, the review aims to assess the clinical implications of vitamin D deficiency in MASLD patients, including its impact on disease severity and progression. By synthesizing existing evidence from preclinical and clinical studies, the review seeks to provide molecular insights into potential therapeutic strategies targeting the vitamin D pathway for the management of MASLD. Moreover, it aims to identify gaps in current knowledge and propose directions for future research to further elucidate the role of vitamin D in MASLD pathophysiology. Ultimately, the goal of the review is to delineate the molecular mechanisms underlying vitamin D-VDR signaling in the context of MASLD development and progression with the overarching objective of improving patient outcomes and guiding the development of effective interventions for this increasingly prevalent liver disease.

## 2. Epidemiology of MASLD

Obesity rates have risen sharply, with the World Health Organization (WHO) estimating that, in 2022, 45% of adults were either obese or overweight [17]. The prevalence of MASLD aligns with this upward trend in obesity, emerging as a significant contributor to chronic liver disease on a global scale. A recent meta-analysis revealed an overall worldwide prevalence of 30%, marking a 50% increase between 1990 and 2019 [18]. Country-specific rates have been documented as 33% in Southeast Asia and South Asia, 44% in Latin America, and 25% in Western Europe [18]. However, the real prevalence is probably underestimated due to the past practice of diagnosing cryptogenic cirrhosis in the absence of viral-, alcohol-, or immune-related liver disease. This increasing prevalence is further underscored by a recent analysis of the European Liver Transplant Registry, which indicated a rise in the percentage of transplant recipients with MASH-related complications from 1.2% in 2002 to 8.4% in 2016 [19].

## 3. Pathophysiology of MASLD

Excessive accumulation of lipids associated with insulin resistance in the absence of viral hepatitis, significant alcohol intake, and other secondary factors are the main characteristics of MASLD [20]. There exists a complex and robust interplay between MASLD and metabolic syndrome, defined by the amalgamation of insulin resistance/type 2 diabetes mellitus (Τ2DM), hypertension, dyslipidemia, and obesity [20]. In a study of individuals with MASH, the obesity prevalence was noted at 82%, Τ2DM at 47%, hyperlipidemia at 72%, and metabolic syndrome at 71% [21]. Τ2DM in MASLD has been highlighted as a risk factor for progressing to MASH, cirrhosis, and mortality [22,23]. Poor glycemic control is related to an increased likelihood of MASH and/or advanced fibrosis [22,23].

The underlying pathological mechanisms of MASLD are based on the “multiple hit” hypothesis, proposing that individuals that are genetically predisposed are influenced by various environmental factors (such as gut microbiome, lifestyle, daily dietary, and the presence of obesity) [22,24]. These factors contribute to lipid accumulation, insulin resistance, obesity, and unfavorable modifications of the gut microbiota [22,25]. Consequently, there is an increase in hepatic de novo lipogenesis and a compromised suppression of adipose tissue lipolysis, leading to an aberrant transportation of free fatty acids to the hepatic tissue and subsequent hepatic fat accumulation. The ensuing lipotoxicity contributes to mitochondrial abnormal function, generating reactive oxygen species (ROS) and inducing endoplasmic reticulum stress (ERS). Insulin resistance and the exaggerated influx of lipopolysaccharides (LPS) from the Intestinal lumen due to gut barrier dysfunction further exacerbate this condition. [25,26]. Injury of the cellular tissue stimulates the infiltration of immune cells, induces fibrogenesis, and encourages subsequent activation of hepatic progenitor cells. Insulin resistance further leads to dysfunction in adipose tissue through the release of pro-inflammatory cytokines and adipokines, intensifying inflammation [22]. At the cellular level, these processes induce oxidative stress and DNA damage in liver tissue, which, in conjunction with inflammation, may culminate in liver fibrosis and cirrhosis [27].

A complete understanding of MASLD natural course remains elusive. The presence of chronic inflammation in the liver seems to play a pivotal role in the transitioning of simple steatosis to MASH and even to HCC [28]. However, the progression from simple steatosis to more severe forms of MASLD does not follow a linear, unidirectional association. Studies involving paired biopsies have revealed a dynamic interplay between MASLD and MASH over time. In a single-center study covering 108 patients over a median of 6.7 years, 42% exhibited signs of progression, 40% showed stable disease, and 18% demonstrated regression [29].

## 4. Vitamin D Physiology and Metabolism

Vitamin D, a fat-soluble vitamin, plays a crucial role in various physiological processes within the human body. Its primary function is the regulation of calcium and phosphorus metabolism, essential for maintaining bone health [30]. In parallel, vitamin D plays a critical role in immune system modulation, cell growth, and differentiation [31,32]. The physiological actions of vitamin D are orchestrated through a series of metabolic conversions that take place primarily in the skin, liver, and kidneys [30] (Figure 1).

The active form of vitamin D is 1,25-dihydroxyvitamin D3 [1,25(OH)2D3], also known as calcitriol [31,32] (Figure 1). There are two major forms of vitamin D with modified chemical structures: vitamin D2, or ergocalciferol, and vitamin D3, or cholecalciferol [33,34]. Even though calcitriol is the active form, the assessment of vitamin D status commonly relies on measuring serum levels of 25(OH)D. This preference is attributed to its comparatively extended half-life and the generally consistent levels observed in the bloodstream. The regulation of vitamin D metabolism is finely tuned using factors such as parathyroid hormone (PTH), calcium and phosphorus levels, and fibroblast growth factor (FGF) [33,35]. The synthesis of 1,25(OH)2D3 is performed not only in renal tissue but also in extra-renal tissues, such as immune cells and intestinal macrophages [36] (Figure 1). Beyond the role of vitamin D in immunomodulation, recent evidence has revealed the great impact of vitamin D on hepatic pathophysiology, revealing its potential to impede fibrosis in liver tissue [37,38]. Numerous studies have established a link between vitamin D insufficiency/deficiency and liver dysfunction, illustrating a negative correlation between 25(OH)D levels and the severity of liver disease [34,38,39,40,41,42,43,44,45,46,47,48].

## 5. Vitamin D Receptor (VDR)

Calcitriol exerts its effects through its binding to the VDR, which is present in various tissues throughout the body [49,50] (Figure 1). The VDR is encoded by the *VDR* gene, belongs to the nuclear receptor superfamily, and acts as a transcription factor, influencing the expression of genes involved in various biological processes in response to vitamin D [51].

Once activated, the VDR forms a complex with the retinoid X receptor (RXR), and this complex binds to specific DNA sequences known as vitamin D response elements (VDREs) located in the promoter regions of target genes [52] (Figure 1). Through this process, the VDR plays a pivotal role in the transcriptional regulation (activation or repression) of various target genes through vitamin-D-mediated signaling or via heterogeneous mechanisms, including interactions with several transcription factors [53] (Figure 1).

The distribution of the VDR is widespread throughout different tissues, including the intestine, kidney, bones, and immune cells, underscoring its involvement in diverse physiological processes [54,55] (Figure 1). Specifically, the VDR is expressed in activated T cells, dendritic cells (DCs), B cells, cytotoxic natural killer (NK) cells, and other antigen-presenting cells (APCs) such as monocytes and macrophages [54,55,56,57]. The presence of inflammatory responses has been linked to the VDR in various immune cell types [58]. The vitamin D-VDR complex displays a suppressive activity on the maturation of DCs; this effect leads to an impaired phenotype that is resistant to maturation stimuli [59]. Concurrently, the vitamin D-VDR complex is involved in T cell receptor (TCR) signaling and T cell activation through the p38 mitogen-activated protein kinase (MAPK) signaling [60]; through this signaling cascade, an inhibitory effect is exerted during the differentiation of CD4+ T helper cells into T helper 1 (Th1) cells, thereby suppressing the effector functions of the latter [61]. In parallel, an activation signal on the expression of T regulatory cells (Tregs) and the release of specific cytokines [interleukin 4 (IL-4), IL-10, and transforming growth factor beta (TGF)-β] occur [61]. VDR expression in the liver displays considerable variation across different cell types and disease stages. In normal hepatocytes, the VDR has notably low expression, while it registers higher levels in non-parenchymal cells, including Kupffer cells (KCs), hepatic sinusoidal endothelial cells, cholangiocytes, and hepatic stellate cells [55]. Both in animal models [62] and human studies [14], VDR expression in the liver is upregulated during the early stages of MASLD and gradually diminishes in the MASH stage [63]. This suggests that conflicting data regarding the role of vitamin D in MASLD may be attributed to differences in the temporal and spatial distribution of VDR expression in hepatic cells.

Dysregulation of the vitamin D-VDR signaling pathway has been associated with various health conditions, including autoimmune diseases, cancer, and metabolic disorders. As research progresses, a deeper understanding of the vitamin D-VDR-related mechanisms and the intricate role of this complex in human physiology continues to emerge, offering potential insights for therapeutic interventions in a range of diseases. Novel findings indicate that vitamin D participates in signaling pathways, regulating the expression of genes with antiproliferative, anti-inflammatory, and anti-fibrotic properties. As a result, vitamin D assumes a crucial role in the progression of chronic liver diseases [34,53,64]. Concurrently, the vitamin D-VDR axis emerges as a pivotal regulator, influencing the development and severity of several liver diseases [34,53,64,65]. More specifically, increasing evidence indicates a strong link between vitamin D deficiency and MASLD—MASH [13,14,15,16]. This association is underscored by the inherent connection of MASLD with obesity and metabolic syndrome.

## 6. Mechanistic Insights between Vitamin D-VDR in MASLD

There is accumulating evidence on the impact of vitamin D on insulin resistance [66,67,68,69] and glucose metabolism [70,71] in individuals with MASLD; however, although data from both in vivo and in vitro studies suggest an association between vitamin D deficiency and MASLD, establishing causality remains elusive. Examining the mechanistic interplay between vitamin D and its receptor provides valuable insights into their role in MASLD. Intra-hepatic and extra-hepatic mechanisms pertaining to the interaction between vitamin D and its receptor, exploring their involvement in the pathogenesis, and progression of MASLD will be thoroughly summarized and discussed below (Figure 2 and Figure 3).

### 6.1. Vitamin D-VDR Mediated Mechanisms in Liver Homeostasis

Vitamin D exhibits anti-inflammatory properties both systemically and in specific tissues, including the liver, as evidenced by experimental studies. In rat models of MASLD, the administration of active vitamin D demonstrated efficacy in mitigating liver inflammation and oxidative stress. This effect was achieved by inhibiting the p53-p21 signaling pathway, thereby impeding associated cell senescence [72] (Figure 2). Additionally, vitamin D exhibited protective properties against high-fat diet (HFD)-induced fatty liver by promoting the nuclear translocation of the antioxidant molecule nuclear factor erythroid 2-related factor 2 (NFE2L2) [73] (Figure 2). Furthermore, vitamin D contributed to the reduction of TLRs [66] and the repression of sirtuin [74] in this context. Vitamin D deficiency deteriorated MASLD progression through the activation of toll-like receptor 2 (TLR2) and TLR4 through the CD14/human LPS-binding protein (LBP), subsequently triggering downstream inflammatory signaling molecules that result in steatosis and inflammation [66] (Figure 2). In addition, vitamin D deficiency induced the activation of hepatic resistin, which potentially plays a role in hepatic signaling alterations and the development of insulin resistance [66]. Increased expression of TLR-4, nuclear factor kappa-light-chain-enhancer of activated B cells (NF-κB), and downstream inflammatory mediators were correlated with diminished vitamin D levels in an in vitro model of primary hepatocytes [75] (Figure 2). This effect was subsequently reversed via vitamin D supplementation, achieved through the downregulation of TLR4-mediated inflammatory pathways [75]. 

Vitamin D demonstrates anti-fibrotic effects in the liver through the inhibition of hepatic stellate cell proliferation and the downregulation of pro-fibrotic mediators, including platelet-derived growth factor (PDGF) and TGF-β (Figure 2). Likewise, vitamin D hinders the expression of collagen, α-smooth muscle actin, and tissue inhibitors of metalloproteinase-1 [76,77] (Figure 2). The protective impact of vitamin D supplementation against HFD-induced MASLD via activation of the VDR has also been described [78]. In this context, VDR knockout mice exhibited exacerbated liver steatosis induced by HFD or methionine- and choline-deficiency [78]. Through mechanistic experiments, it was discovered that the VDR interacts with hepatocyte nuclear factor 4α (HNF4α) [78]. Overexpression of HNF4α was found to ameliorate HFD-induced MASLD and metabolic abnormalities in mice with liver-specific VDR knockout, suggesting that vitamin D improves MASLD and metabolic irregularities by activating the hepatic VDR, fostering its interaction with HNF4α [78] (Figure 2).

Contradictory results were presented by an investigation demonstrating that VDR deletion protected against liver steatosis, dyslipidemia, and insulin resistance in apolipoprotein E (apoE) knockout (−/−) HFD mice [67]. These mice also displayed reduced synthesis of taurine-conjugated bile acids. In parallel, VDR deletion in apoE−/− mice on HFD induced the inhibition of lipogenesis-related genes (*CD36*, *DGAT2*, *C/EBPα*, and *FGF21*) and the activation of fat oxidation genes (*PNPLA2*, *LIPIN1*, and *PGC1α*) (Figure 2). Exposure of HFD apoE−/− mice to paricalcitol, a non-calcemic vitamin D analogue, resulted in the early induction of hepatic VDR expression in the context of a fatty liver [67].

Vitamin D improves adipose tissue inflammation and protects against liver steatosis by decreasing both the release of lipid droplets from adipose tissue and the hepatic processes of de novo lipogenesis and fatty acid oxidation [79]. Additionally, the administration of calcitriol enhances the expression of the VDR in peripheral cells, thereby improving the overall inflammatory profile at the systemic and tissue levels. This leads to a reduction in adipose tissue inflammation and liver steatosis in animal models [80]. A potential biochemical mechanism by which supplementation of vitamin D modulates glucose metabolism in diabetes has been described. Specifically, vitamin D treatment is proposed to induce the transcriptional regulation and translocation of glucose transporter 4 (GLUT4) in adipocytes and reduce the expression of inflammatory markers, thereby augmenting glucose uptake and utilization [70] (Figure 3A).

Jahn et al. proposed that the activation of the VDR in intestinal tissue led to suppressed *ANGPTL4* expression [81] (Figure 3B). This process resulted in pro-adipogenic/steatogenic effects in adipose tissue and the liver, suggesting a VDR-mediated metabolic crosstalk between intestinal and adipose tissue, which greatly contributes to systemic lipid homeostasis [81].

In contrast, García-Monzón et al. documented an elevation in hepatic angiopoietin-like 8 (ANGPTL8) expression in individuals with MASLD. Moreover, the mRNA levels of *ANGPTL8* were found to correlate with both *VDR* mRNA and the extent of liver steatosis [82]. A mechanistic approach from the same study proposed that *ANGPTL8* is a newly identified target gene of the VDR, playing a role in fostering triglyceride accumulation in hepatocytes [82] (Figure 2).

### 6.2. Autophagy-Related Mechanisms Mediated via Vitamin D-VDR

Autophagy is triggered by TLR-mediated signaling. TLRs are located at the cell surface and endosomes. TLR-mediated signaling induces the transcription of genes implicated in T cell activation, inflammatory, and antiviral responses [83]. During autophagy, abnormally folded proteins, malfunctioning macromolecules (glycogen, lipids, and nucleic acids), and defective organelles are removed via lysosomal clearance [84,85]. The impaired modulation of autophagy may be a contributing factor to the pathogenesis of various human diseases, such as liver-associated diseases, neurodegenerative diseases, and tumorigenesis [86].

In the context of MASLD, disrupted autophagy becomes apparent. Factors such as obesity or prolonged HFD exposure can compromise the cellular autophagic machinery at different stages. This includes the inhibition of autophagosome formation, hindrance in the fusion of autophagosomes and lysosomes, and alterations in lysosomal physiology [87,88]. Additionally, the accumulation of lipids can modify the composition of autophagosome membranes, leading to an impaired fusion process between lysosomes and autophagosomes [89]. Knockdown of autophagy-related genes or their pharmacological inhibition may result in the accumulation of triglycerides in hepatocytes [87,89,90]. Several studies have reported compromised autophagy in both clinical [91] and experimental [87,92] settings.

The impact of vitamin D on autophagy remains a subject of controversy; however, various studies argue that vitamin D is able to enhance autophagy and modulate biological processes in different organs through genomic and non-genomic actions [93,94]. In parallel, VDR signaling has a major impact on autophagy via the modulation of inflammation and host immunity by activating antimicrobial defenses [95,96]. The role of vitamin D in safeguarding against oxidative stress by modulating autophagy has been explored in prior studies, investigating scenarios such as hepatic ischemia–reperfusion in mice [95] and hepatic steatosis [97]. In the case of hepatic ischemia–reperfusion, an augmented autophagic flux was observed following vitamin D administration, which was evidenced by an increase in protein light chain 3 (LC3) conversion both in vivo and in vitro [95]. The crucial involvement of both MEK/ERK and PTEN/PI3K/Akt/mTOR pathways in vitamin-D-induced autophagy has been established [95]. Moreover, inhibition of either the MEK/ERK or PTEN/PI3K/Akt/mTOR pathways partially nullified the protective effect of vitamin-D-induced autophagy [95]. Furthermore, the initiation of autophagy signaling pathways was hindered by the knockdown of Beclin-1, completely reversing the protection conferred by vitamin D [95]. Vitamin D has also been found to improve hepatic steatosis by enhancing autophagy via the upregulation of autophagy-related 16-like 1 (ATG16L1) [97].

Specifically, the liver damage and steatosis induced by a HFD in mice were mitigated via treatment with 1,25(OH)2D3. This was accompanied via an augmentation of autophagy and an upregulation of ATG16L1 expression [97]. The protective effects of 1,25(OH)2D3 on hepatic steatosis were impeded when autophagy-induced by 1,25(OH)2D3 was inhibited by 3-methyladenine (3-MA) [97]. Moreover, the 1,25(OH)2D3-induced autophagy contributed to anti-inflammatory events and the modulation of lipid metabolism in the liver [97]. In HepG2 cells, vitamin D decreased the accumulation of lipids while promoting autophagy and ATG16L1 expression. However, this effect was nullified after VDR knockdown [97]. These findings highlighted the potential use of vitamin D in conjunction with autophagy activation for the management of MASLD [98]. Lim et al. presented that vitamin D3 treatment activated autophagy by inducing AMP-activated protein kinase (AMPK), inhibiting Akt, and ultimately suppressing the activation of the mammalian target of rapamycin (mTOR) in T2DM mice, thus ameliorating impaired hepatic lipid regulation [99]. Moreover, vitamin D3 demonstrated positive effects on anti-apoptotic and anti-fibrotic activities in the liver of diabetic mice [99]. Calcitriol treatment in HFD mice with MASH led to a reduction in p62, an autophagic cargo adaptor, suggesting the induction of autophagy [100]. This study proposed a potential mechanism for suppressing the inflammatory response that occurs during lipid accumulation in MASH. The proposed mechanism involves reducing the overactivation of the NOD-like receptor protein 3 (NLRP3) inflammasome [100]. The observed attenuation of this effect with simultaneous administration of the autophagy inhibitor, hydroxychloroquine, implies that the inhibitory impact of calcitriol on NLRP3 inflammasomes and pyroptosis may be dependent on autophagy [100].

### 6.3. ERS-Related Mechanisms Mediated via Vitamin D-VDR

Sustained and chronic ERS, especially in the liver, can result in apoptosis, oxidative injury, inflammation, and steatosis [101]. Maintaining ER homeostasis in the liver is crucial for controlling intra- and extra-hepatic lipid content. Increased evidence has emphasized the role of ERS in the pathogenesis of MASLD and MASH [102,103]. ERS and autophagy are the key players in the process of hepatic ischemia–reperfusion injury [104]. In this setting, the potential of vitamin D to regulate ERS in an autophagy-dependent manner has been described. In macrophages, vitamin D can inhibit ERS and activate the adaptive unfolded protein response (UPR) to restore homeostasis [105]. In contrast, vitamin D deficiency results in persistent, unresolved ERS; deteriorating liver injury; and inflammation. In the case of hepatic injury, markers of ERS and apoptosis, such as IRE1a and CHOP, respectively, were found to hinder the anti-inflammatory M2 macrophage polarization phenotype, leading to increased inflammatory responses in hepatocytes [106,107]. Previous studies have indicated that activation of the VDR in both liver and macrophage cell lines resulted in decreased levels of CHOP, NF-κB p65, and inflammatory cytokines, whereas vitamin D deficiency diminished these beneficial effects [108].

### 6.4. Microbiome-Related Mechanisms Mediated via Vitamin D-VDR

The intestinal microflora stands as a pivotal factor in the progression of MASLD. In parallel, vitamin D deficiency has been strongly related to the intestinal microbiome by impacting nutrient absorption from the diet and the enterohepatic cycle of bile acids [109]. Increased consumption of fat and carbohydrates has been demonstrated to elevate circulating LPS, contributing to significant hepatic insulin resistance [110]. LPS plays a critical role in activating the immune system, ultimately leading to systemic inflammation and obesity [111]. Dysbiosis or changes in gut microbiota composition can result in heightened intestinal permeability and chronic inflammation in individuals with obesity, metabolic syndrome, and MASLD [112,113].

Vitamin D has been found to modify the distribution of the fecal microbiome; elevated vitamin D levels have been related to increased levels of beneficial bacteria and decreased levels of pathogenic genera [114,115]. Deletion of the *VDR* gene in experimental models has been found to affect the gut microbiome at both structural and functional levels. This alteration heightened the susceptibility to infection, inflammation, cancer, and various other diseases [116]. Likewise, in preclinical models of MASLD, a vitamin-D-deficient HFD exacerbated endotoxemia, gut permeability, dysbiosis, inflammation, insulin resistance, and steatosis of the liver, while vitamin D supplementation demonstrated a mitigating effect on steatosis [68] (Figure 3B). The VDR significantly contributes to the maintenance of gut homeostasis. The increased expression of the VDR in the digestive system underscores its role in maintaining gut homeostasis by engaging in a range of regulatory activities [117,118]. These activities contribute to the preservation of intestinal barrier function, immunomodulation, and regulation of the gut microbiota [68,78,119,120,121]. Notably, the VDR was found to govern the expression of tight junction proteins (claudin 2 and 12) by upregulating them and downregulating cadherin-17, thereby influencing the integrity of the intestinal barrier [122,123] (Figure 3B). Additionally, the VDR plays a crucial role in hindering intestinal apoptosis, thereby controlling gastrointestinal inflammation [63].

The amelioration of MASH in mice has been described in a study that used a lipid-based nanocarrier containing active vitamin D3 [124]. The use of vitamin D nanocarriers inhibited the intestinal permeability of an epithelial layer induced via LPS in vitro [124] (Figure 3B). In vivo, oral administration of nanocarriers improved the heightened permeability in the intestinal barrier and reduced steatosis, inflammation, and fibrosis in the liver [124] (Figure 3B).

The abundant VDR expression in the distal region of the small intestine, particularly in areas rich in Paneth cells, suggests a potential role for vitamin D signaling in the modulation of intestinal Paneth cells. This modulation could influence the production of defensins, thus controlling the growth of the microbiome in the small intestine [125]. Enhanced gut permeability results in bacterial translocation, which leads to the activation of TLRs on KCs [126]. Vitamin D mitigates the expression of TLRs in KCs, thereby diminishing inflammation [126] (Figure 2). Dysfunction in the vitamin D-VDR axis may compromise innate immunity in the gut, potentially leading to liver pathology. This compromise involves the downregulation of alpha defensins via Paneth cells, bacterial translocation, endotoxemia, low-grade inflammation, insulin resistance, and hepatic steatosis [69].

Wu et al. presented the significance of vitamin D hepatic 25-hydroxylation in the small intestine for maintaining Paneth cell function [125]. In the context of carbon-tetrachloride-induced liver injury, there deficient hepatic 25-hydroxylation of vitamin D was observed, resulting in the downregulation of defensins expressed via Paneth cells [125]. This deficiency resulted in gut dysbiosis and endotoxemia [125] (Figure 3B). Intestinal VDR knockout mice exhibited reduced Paneth cell defensins and lysozyme production, exacerbating liver injury and fibrogenesis [125]. These findings suggest that liver injury hampers vitamin D synthesis, disrupting Paneth cell function in the small intestine and triggering gut dysbiosis and liver fibrogenesis [125]. Given the critical role of the gut microbiota in MASLD progression, an analysis of the gut microbiota in MASLD rats revealed that vitamin D reversed HFD-induced dysbiosis by increasing *Lactobacillus* and decreasing *Acetatifactor*, *Oscillibacter*, and *Flavonifractor* [127]. This study implies that vitamin D might alleviate HFD-induced MASLD by modifying the gut microbiota composition [127]. In a recent integrated microbiota and metabolomic analysis, the same research team revealed that treatment with vitamin D suppressed the HFD-induced MASLD and restored the gut microbiota and metabolism dysbiosis, proposing the potential of vitamin D supplementation as an alternative intervention for MASLD by targeting specific microbiota populations [128]. In particular, vitamin D treatment increased *Prevotella* abundance, which was positively correlated with serotonin, melatonin, tryptamine, L-arginine, and 3-dehydrosphinganine [128]. The decrease in *Mucispirillum* after vitamin D supplementation maintained a positive correlation with plasma tryptophan, tyrosine, glutamic acid, phenylalanine, branched chain amino acids, sphinganine, and spermidine [128]. These data suggest that vitamin D can enhance tryptophan and tyrosine metabolism and arginine biosynthesis by inhibiting *Mucispirillum* proliferation [128]. Additionally, vitamin D treatment promoted sphingolipid metabolism in both gut microbiota function and metabolite pathways [128].

### 6.5. MiRNA-Related Mechanisms Mediated via Vitamin D-VDR

MicroRNAs (miRNAs), a class of small, single-stranded, non-coding ribonucleic acid (RNA) molecules of 19 to 24 nucleotides, are expressed in most eukaryotes, including humans [129]. MiRNAs play a crucial role in the regulation of gene expression by targeting messenger RNA (mRNA), leading to post-transcriptional gene silencing or mRNA degradation, ultimately influencing protein production [130]. They actively participate in a range of biological functions, including cell proliferation, maturation, and differentiation; signal transduction; cell apoptosis; modulation of chronic inflammation; and carcinogenesis [131,132]. Additionally, these molecules have been implicated in the regulation of diverse cellular and metabolic pathways [131]. Impaired regulation of miRNAs occurs in various pathological conditions, including metabolic dysfunction [133] and liver-associated diseases [134]. Notably, miRNA-regulated pathways have been implicated in the progression of MASLD to MASH, indicating their potential as diagnostic and therapeutic tools in this context [135,136].

In the MASLD setting, there are dysregulated miRNAs which have the potential to modulate transcription factors associated with hepatic lipogenesis and lipid or carbohydrate metabolism, such as carbohydrate response element-binding protein (ChREBP), sterol regulatory element-binding protein 1 (SREBP-1), and peroxisome proliferator-activated receptors (PPARs) [137]. The cellular response to vitamin D is affected by miRNAs through various mechanisms (mainly via VDR post-transcriptional regulation) [138]. The interaction between vitamin D and miRNAs is bidirectional. MiRNAs can influence not only vitamin D metabolism but also the vitamin D-VDR signaling pathway by regulating various genes, including CYP27B1, CYP24A1, and RXRα [139,140]. Conversely, the transcription of various miRNAs is directly modulated by VDR signaling [141]. Although there is evidence describing the association between miRNA activity and vitamin D status in several liver-related pathologies, there is a lack of data in the context of the possible contributions of vitamin D to miRNAs, in MASLD pathogenesis.

Given the prominent impact of vitamin D on inflammation and lipid metabolism, there is a hypothesis that supplementation with vitamin D may regulate fibrogenic miRNAs in individuals with MASLD. Ebrahimpour-Koujan et al. designed an ongoing randomized clinical trial aiming to determine the impact of vitamin D supplementation on VDR serum levels, and fibrogenic miRNAs (miR-21, miR-34, and miR-122) in patients with MASH [142]. A recent study using Venn analysis revealed a number of vitamin-D-related miRNAs, including miR-27, miR-125, miR-155, miR-192, miR-223, miR-375, and miR-378, which may contribute to MASLD pathogenesis [143]. In particular, the miR-27b, which targets and regulates the VDR [144], has shown modified expression in the serum of MASLD patients [145,146,147]. In parallel, miR-192 was found to be reduced in the serum of patients with prediabetes under vitamin D supplementation (2000 IU cholecalciferol) for 4 months, correlating with favorable alterations in glucose levels and the disposition index [71].

A recent study by Tanoglu et al. [148], using a rat model of high-fructose-diet-induced fatty liver, revealed insights into the relationship between vitamin D supplementation and the expression and circulation of microRNAs 200c and 33a. The findings of the study clarified that vitamin D plays a protective role against MASLD, and there was a notable alteration in the circulating levels of miR-200c and 33a associated with vitamin D. Specifically, the results revealed more favorable histopathological alterations in the vitamin-D3-supplemented group including attenuating inflammatory, metabolic, and hepatic effects through miR-33a and miR-200c [148]. Moreover, within a specific dosage and a defined time frame, vitamin D3 showed the capacity to reduce hepatic lipid accumulation, inflammation, and fatty degeneration induced by high-fructose corn syrup [148]. These data reinforce the hypothesis that circulating miRNA levels could serve as biomarkers for MASLD [148].

### 6.6. Inflammasome-Related Mechanisms Mediated via Vitamin D-VDR

The NLRP3 inflammasome is a major component of the immune system’s response to tissue injury, infection, and cellular stress [149,150]. The NLRP3 inflammasome plays a critical role in the regulation of inflammation by controlling the stimulation of pro-inflammatory cytokines [151]. The activation of the inflammasome, along with its key components such as the sensor protein, adaptor protein ASC, and caspase-1, contributes to the inflammatory processes and immune responses associated with MASLD, thereby influencing the overall pathophysiology of the liver condition [152]. Specifically, the excessive activation of NLRP3 has been identified as a potential driver in the progression of MASLD towards MASH [153]. This activation is linked to inflammasome-mediated cell death, commonly referred to as ‘pyroptosis’ [154]. Elevated levels of NLRP3, ASC, and caspase-1 in the liver have been associated with liver inflammation in the context of MASLD [155,156].

Zhang et al. demonstrated that vitamin D treatment effectively suppressed the overactivation of the NLRP3 inflammasome and pyroptosis induced by MASLD, thereby alleviating hepatic injury caused by a HFD [127]. This protective effect was observed both in vivo and in vitro, where vitamin D treatment rescued the compromised cell viability of BRL 3A fibroblast-like cells isolated from rat livers, subjected to treatment with palmitic acid and LPS [127]. In liver tissue samples, this protective effect was substantiated by the downregulation of inflammasome components and cytokines, including NLRP3, ASC, caspase-1, IL-1β, IL-18, and gasdermin D [127] (Figure 2).

A recent study investigated whether the improvement of MASH via calcitriol could be attributed to the inhibition of the overactivated NLRP3 inflammasome [100]. The results showed that calcitriol treatment effectively counteracted the histopathological abnormalities observed in the liver of HFD mice, coinciding with the restoration of *VDR* gene expression and a decrease in sterol regulatory element binding protein 1c (SREBP1c) levels [100]. Furthermore, calcitriol resulted in decreased levels of NLRP3, caspase-1, gasdermin D, and IL-18 proteins, along with a decrease in *ASC* gene expression and reduced IL-1β and caspase-3 immunoreactivities, compared to the control group [100].

### 6.7. Impact of Vitamin D-Associated Genetic Variants in MASLD

Multiple genetic polymorphisms of the *VDR* gene or vitamin-D-associated genes have been linked to both the occurrence and severity of MASLD, potentially impacting the regulation of adipose tissue function and fibrosis. Vitamin D binds to the hepatic VDR, exerting its biological functions by either activating VDR transcriptional activity to regulate the expression of genes associated with inflammation and fibrosis or by initiating intracellular signal transduction through VDR-mediated activation of Ca^2+^ channels. In our recent review, we detailed the combined impact of *VDR* gene polymorphisms along with vitamin D-VDR signaling on the development and progression of MASLD [157].

## 7. Vitamin D Supplementation in MASLD Patients

Studies have reported conflicting results related to vitamin D supplementation in patients with MASLD or MASH. Vitamin D administration has been proven beneficial in various studies [158,159,160]; however, there are reports of low or no beneficial effect in the same setting [161,162,163]. The disparities presented in study outcomes primarily stem from variations in individuals’ baseline vitamin D levels at the study’s outset, emphasizing the need to standardize vitamin D assessment to ensure consistent baseline levels among participants. It is crucial to acknowledge that uniform results from the same supplementation regimen are unlikely for individuals with differing circulating vitamin D levels. Additionally, standardizing vitamin D measurement is essential to accurately define participants’ vitamin D status upon enrollment in trials. Moreover, the dosing regimens utilized in these studies play a vital role in the determination of the overall balance of vitamin D, consequently impacting its therapeutic outcomes on organs and systems. Research in both children [164] and adults [165] has demonstrated that administering vitamin D loading doses, such as monthly or weekly boluses, is the most effective approach for rapidly increasing circulating 25-hydroxyvitamin D3 concentrations, leading to potentially optimal levels sooner than daily supplementation [77]. Moreover, consideration should be given to inter-individual differences in vitamin D metabolism, influenced by various factors including body fat levels, intestinal absorption efficiency, and genetic polymorphisms [166]. Additionally, different forms of vitamin D supplements exhibit varying efficacies in improving circulating 25(OH)D3 levels, with cholecalciferol demonstrating more advantages compared to ergocalciferol.

Another important point is the study of the long-term kinetics of vitamin D supplementation effects and monitoring over time in patients with MASLD to understand the sustained impact of supplementation on disease progression, liver health, and overall metabolic status. This longitudinal approach allows for the assessment of how vitamin D levels fluctuate over time, how they correlate with changes in liver function and metabolic parameters, and whether sustained supplementation leads to durable improvements in MASLD outcomes. Additionally, long-term monitoring can help identify any potential adverse effects or risks associated with prolonged vitamin D supplementation in this patient population.

## 8. Conclusions

Ongoing research conducted in recent decades has emphasized the pivotal role of vitamin D in regulating liver homeostasis, inflammation, and fibrogenesis. Vitamin D actively influences signaling pathways that govern the expression of genes associated with antiproliferative, anti-inflammatory, and anti-fibrotic effects. In the clinical setting, the deficiency of vitamin D and the presence of genetic variants in vitamin-D-related genes are closely associated with the development or progression of liver diseases. These findings underscore the clinical relevance of vitamin D in influencing key processes and pathways associated with liver homeostasis and disease progression. MASLD emerges as a complex, multifaceted condition intricately linked to various factors, including oxidative stress, insulin resistance, lipid abnormalities, metabolic dysregulation, obesity, immune modulation, and alterations in the gut microbiota. In the context of MASLD, vitamin D deficiency is commonly associated with disease existence and is greatly correlated with disease severity. Experimental evidence supports the notion that vitamin D deficiency contributes to metabolic deregulation, whereas vitamin D per se exhibits anti-inflammatory properties in hepatic cells. The vitamin D-VDR axis plays a direct role in modulating metabolic and inflammatory pathways linked to the progression of MASLD in overweight and obese individuals. Notably, the vitamin D-VDR pathway influences hepatic processes such as lipogenesis and bile acid circulation within hepatocytes. Additionally, this pathway has the potential to improve the control of hepatic inflammation by regulating the polarization of hepatic macrophages, maintaining immune homeostasis through the regulation of other hepatic immune cells, and inhibiting liver fibrosis by impeding hepatic stellate cell activation. The vitamin D-VDR axis contributes not only to intra-hepatic regulation but also plays a crucial role in maintaining homeostasis in organs pivotal to the pathogenesis of MASLD and MASH, such as the gut and adipose tissue. In parallel, studies suggest the influence of vitamin D on the expression and regulation of miRNAs in the context of liver diseases, potentially impacting pathways involved in metabolic regulation, inflammation, and fibrosis. Understanding the intricate relationship between vitamin D and miRNAs in MASLD could unveil novel insights into the molecular mechanisms underlying the disease. Thus, further research in this setting may contribute to the development of targeted therapeutic strategies that involve both vitamin D and miRNA modulation for the management of MASLD.

Vitamin D supplementation holds a potential therapeutic avenue for MASLD and MASH by improving prognosis at both biochemical and histological levels, owing to its pleiotropic effects that extend across immunological, hormonal, metabolic, and inflammatory activities. However, it is emphasized that more extensive clinical trials are essential to directly assess the efficacy of vitamin D supplementation on the factors driving MASLD progression, especially in vitamin D deficient individuals. This is crucial for improving our understanding and, consequently, the applicability of the molecular insights derived from experimental studies.

Moving forward, well-defined future studies should concentrate on unraveling the mechanistic and genetic insights associated with vitamin D-VDR in the context of MASLD. This comprehensive understanding will not only contribute to refining treatment strategies but will also shed light on potential preventive measures, ultimately addressing the multifaceted nature of MASLD.

## Figures and Tables

**Figure 1 ijms-25-04901-f001:**
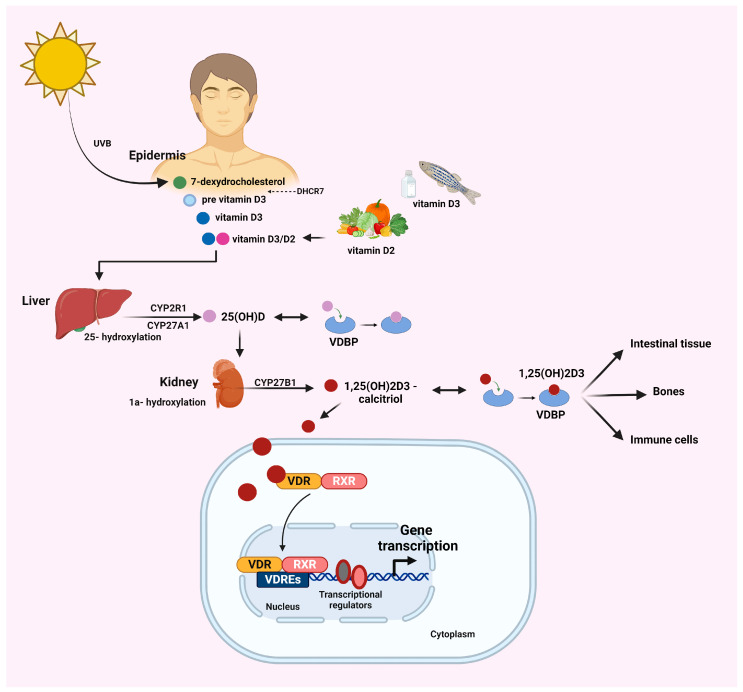
Vitamin D biosynthesis and metabolism and the vitamin D-VDR signaling axis. The synthesis of vitamin D begins in the skin when ultraviolet B (UVB) sunlight exposure triggers the conversion of 7-dehydrocholesterol, a compound found in the skin, into pre-vitamin D3. This pre-vitamin D3 then undergoes a thermal isomerization to form vitamin D3. Alternatively, vitamin D can be obtained from dietary sources such as fatty fish, fortified foods, and supplements; this form is absorbed by the small intestine in conjunction with chylomicrons. Following this, vitamin D is transported to the liver attached to vitamin D binding protein (DBP). Once synthesized or ingested, vitamin D undergoes a two-step activation process. In the liver, it is converted by the enzyme 25-hydroxylase (CYP2R1) into 25-hydroxyvitamin D [25(OH)D], the major circulating form of the vitamin D. Subsequently, in the kidneys, 25(OH)D is further hydroxylated by the enzyme 1α-hydroxylase (CYP27B1) to form the biologically active form of vitamin D, 1,25-dihydroxyvitamin D3 [1,25(OH)2D3]. Upon the activation of vitamin D, a conformational rearrangement of the vitamin D receptor (VDR) takes place, which results in heterodimerization with the retinoid X receptor (RXR). This complex is transferred to the nucleus, binds to specific genomic sequences (VDREs) in the promoter region, and modulates the transcription of target genes. The VDR is expressed in numerous tissues and cells, including the intestinal tissue, kidney, bones, and immune cells [activated T cells, dendritic cells (DCs), B cells, cytotoxic natural killer (NK) cells, and other antigen-presenting cells such as monocytes and macrophages] in response to antigenic challenges. These cells constitute vitamin D targets. Abbreviations: UVB, ultraviolet B; DHCR7, 7-dehydrocholesterol reductase; CYP2R1, cytochrome P450, family 2, subfamily R, polypeptide 1; CYP27A1, cytochrome P450, family 27, subfamily A, polypeptide 1; CYP27B1, cytochrome P450, family 27, subfamily B, polypeptide 1; VDBP, vitamin D binding protein; VDR, vitamin D receptor; RXR, retinoid X receptor; VDRE, vitamin D response element.

**Figure 2 ijms-25-04901-f002:**
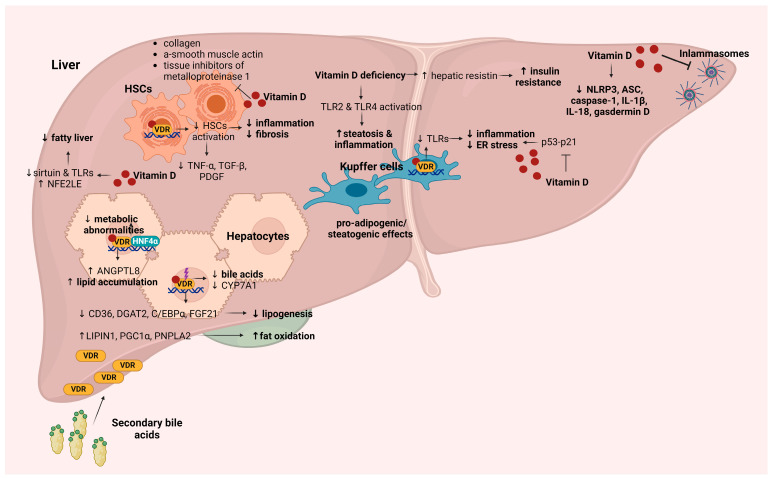
Potential mechanisms of intra-hepatic effects of the vitamin D-VDR axis involved in the development and progression of MASLD. Vitamin D-VDR-mediated mechanisms in liver tissue result in the attenuation of inflammation, impediment of fibrotic effects, reduction of steatosis and insulin resistance, alleviation of endoplasmic reticulum stress and metabolic abnormalities, reduction of lipogenesis, and enhancement of fat oxidation, collectively contributing to improved metabolic health and liver function. Abbreviations: TLR, toll-like receptor; NFE2L2, nuclear factor erythroid 2-related factor 2; HSC, hepatic stellate cells; TNF-α, tumor necrosis factor α; TGF-β, transforming growth factor beta; VDR, vitamin D receptor; HNF4α, hepatocyte nuclear factor 4 α; ER, endoplasmic reticulum; IL-1, interleukin 1; CYP7A1, cholesterol 7 alpha-hydroxylase; ANGPTL8, angiopoietin like 8; CD36, cluster of differentiation 36; DGAT2, diacylglycerol O-acyltransferase 2; C/EBPα, CCAAT/enhancer binding protein α; FGF21, fibroblast growth factor 21; PGC1α, peroxisome-proliferator-activated receptor gamma coactivator 1-alpha; PNPLA2, patatin-like phospholipase domain-containing protein 2.

**Figure 3 ijms-25-04901-f003:**
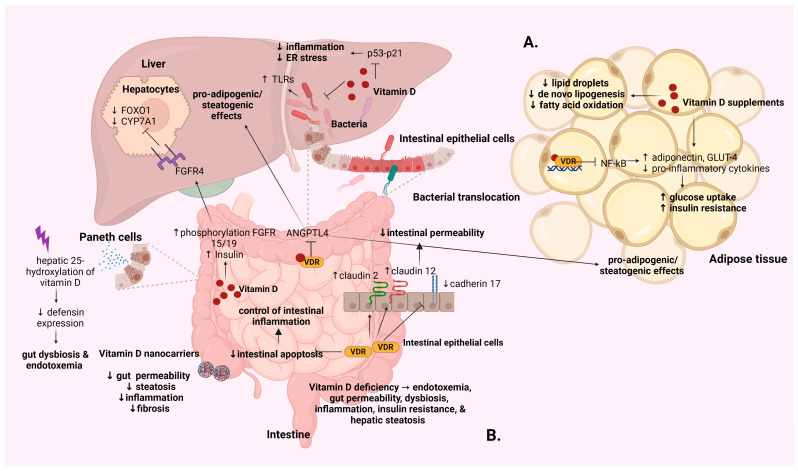
Potential mechanisms of extra-hepatic effects of the vitamin D-VDR axis involved in the development and progression of MASLD. (**A**) Molecular mechanisms by which vitamin D-VDR signaling influences adipose tissue and contributes to MASLD progression include the regulation of adipogenesis, modulation of inflammation, enhancement of insulin sensitivity, regulation of lipid metabolism, and potential mitigation of adipose tissue fibrosis, collectively impacting metabolic dysfunction and liver pathology. (**B**) Vitamin D-VDR mechanisms in intestinal tissue impact MASLD progression through the modulation of intestinal barrier integrity, gut microbiota composition, and inflammatory responses, potentially influencing systemic metabolic homeostasis and liver health. Abbreviations: TLR, toll-like receptor; FOXO1, forkhead transcription factor 1; CYP7A1, cholesterol 7a-hydroxylase; FGFR4, FGF receptor 4; ANGPTL4, angiopoietin like 4; VDR, vitamin D receptor; FGFR15/19, fibroblast growth factor receptor 15/human ortholog 19; NF-κB, nuclear factor-κB; GLUT4, glucose transporter 4.

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
