# Peer review of "Vitamin D and Metabolic Dysfunction-Associated Steatotic Liver Disease (MASLD): Novel Mechanistic Insights"

_ijms, 2024, doi:10.3390/ijms25094901_

Round 1
Reviewer 1 Report
Comments and Suggestions for Authors
In the present review, Aggeletopoulou et al. discuss the protective effects of vitamin D against metabolic dysfunction-associated steatotic liver disease (MASLD). The manuscript is well-written and provides a detailed analysis of vitamin D metabolism and its impact on the MASLD pathophysiology. As such, vitamin D could be a valuable micronutrient to combat this highly prevalent disease. However, this topic has already been discussed in recent reviews (cited in the manuscript), which may decrease the novelty of the present work.
Remarks to the authors:
1. In line 63, please include the full name of VDR.
2. In the last paragraph of the introduction, please explain in a bit more detail what aspects of the vitamin D-MASLD interplay discussed in the present review were not previously discussed in related reviews (such as references 13-15). This additional information would better highlight the novelty of the present study.
3. In lines 67-68, please include a reference for the statement "Obesity rates have significantly augmented with the World Health Organization (WHO) estimating that in 2016, 39% of adults were either obese or overweight".
4. In lines 108-110, please rephrase the sentence: "Pivotal role in the transitioning of simple steatosis to MASH and even to HCC seems to play the presence of chronic inflammation in the liver [26]". It might sound better as "The presence of chronic inflammation in the liver seems to play a pivotal role in the transitioning of simple steatosis to MASH and even to HCC [26]".
5. In the legend of Figure 1, please include a brief description of this figure, briefly summarizing the effects of sunlight on the skin, and the metabolic effects on the liver and kidneys. Please also mention in what organ(s) and cells the gene expression alterations described in this figure occur. To avoid any repetition, please consider moving some information from lines 149-174 to this figure legend.
6. Please consider mentioning Figure 1 in the main text more often, for example, also in the second paragraphs of sections 4 and 5.
7. In lines 228-231, please include references for these statements.
8. In the legend of Figure 2, please include a brief description of this Figure. To avoid any repetition, please consider moving some information from section 6 to this figure legend.
9. Figure 2 is quite complex. Please consider dividing it into panels (possibly one for each organ) and referring to each of its panels in the main text.
10. In line 578, please correct "(...) vitamin D deficiency contribute (....)" to "(...) vitamin D deficiency contributes (....)"
11. Please consider mentioning the following article (DOI: 10.1080/14728222.2023.2274099).
Author Response
Reviewer 1
In the present review, Aggeletopoulou et al. discuss the protective effects of vitamin D against metabolic dysfunction-associated steatotic liver disease (MASLD). The manuscript is well-written and provides a detailed analysis of vitamin D metabolism and its impact on the MASLD pathophysiology. As such, vitamin D could be a valuable micronutrient to combat this highly prevalent disease. However, this topic has already been discussed in recent reviews (cited in the manuscript), which may decrease the novelty of the present work.
Comment 1: In line 63, please include the full name of VDR.
Response to comment 1: We have revised, as suggested (lines 63-64).
Comment 2: In the last paragraph of the introduction, please explain in a bit more detail what aspects of the vitamin D-MASLD interplay discussed in the present review were not previously discussed in related reviews (such as references 13-15). This additional information would better highlight the novelty of the present study.
Response to comment 2: We have elaborated on the Review aim, as suggested (lines 62-76).
Comment 3: In lines 67-68, please include a reference for the statement "Obesity rates have significantly augmented with the World Health Organization (WHO) estimating that in 2016, 39% of adults were either obese or overweight".
Response to comment 3: While the review was under review, WHO announced the data for 2022. Therefore, we updated the data and added the related reference as suggested (lines 80-81).
Comment 4: In lines 108-110, please rephrase the sentence: "Pivotal role in the transitioning of simple steatosis to MASH and even to HCC seems to play the presence of chronic inflammation in the liver [26]". It might sound better as "The presence of chronic inflammation in the liver seems to play a pivotal role in the transitioning of simple steatosis to MASH and even to HCC [26]".
Response to comment 4: We have rephrased the sentence as suggested (lines 123-124).
Comment 5: In the legend of Figure 1, please include a brief description of this figure, briefly summarizing the effects of sunlight on the skin, and the metabolic effects on the liver and kidneys. Please also mention in what organ(s) and cells the gene expression alterations described in this figure occur. To avoid any repetition, please consider moving some information from lines 149-174 to this figure legend.
Response to comment 5: We have revised as suggested. We have elaborated in the figure legend 1 and we have removed the duplicated data from the main text (lines 151-170).
Comment 6: Please consider mentioning Figure 1 in the main text more often, for example, also in the second paragraphs of sections 4 and 5.
Response to comment 6: We have mentioned the figure where necessary, as suggested.
Comment 7: In lines 228-231, please include references for these statements.
Response to comment 7: We have added the relevant references, as suggested (line 247).
Comment 8: In the legend of Figure 2, please include a brief description of this Figure. To avoid any repetition, please consider moving some information from section 6 to this figure legend.
Response to comment 8: Figure 2 involves all the information related to the mechanisms involved in MASLD pathophysiology and the signaling pathways that critically contribute to this process. Thus, a description of the mechanisms presented in this figure in the figure legend may be helpful for the reader; however, we believe that the presentation of these mechanisms in the main text accompanied with the relevant references is necessary. Considering that the detailed description in both legend and main text may lead to repetition, we have added in the figure legend more information on the evidence presented, but not the mechanisms described (lines 261-282).
Comment 9: Figure 2 is quite complex. Please consider dividing it into panels (possibly one for each organ) and referring to each of its panels in the main text.
Response to comment 9: We have divided the figure 2 into 6 compartments in order to be better apprehending, as suggested.
Comment 10: In line 578, please correct "(...) vitamin D deficiency contribute (....)" to "(...) vitamin D deficiency contributes (....)"
Response to comment 10: We have revised, as suggested (line 663).
Comment 11: Please consider mentioning the following article (DOI: 10.1080/14728222.2023.2274099).
Response to comment 11: Τhe reference has been added to the revised manuscript, as suggested.
Reviewer 2 Report
Comments and Suggestions for Authors
The paper by Aggeletopoulou et al provides a quite extensive but somewhat confusing overview on vitamin D and liver disease . A detailed presentation on MASLD epidemiology and pathophysiology can be found today in any textbook and the Authors made no effort in establishing a direct link with Vitamin D intake and epidemiology: more in some parts of the word? Less?
They did a relatively good job in detailing the metabolic fate of vitamin D although again this has little to do with the topic in discussion. The vitamin D receptor is an interesting topic althogh Figure 1 is difficult to follow. It seems to have analogy with the PPAR system (complexed with the RXR). I do not understand whether the lipid changes occurring in MASLD have anything to do with this. Actually an interaction between VDR and the PPAR gamma has been well described apparently in adipose tissue development. I do not see any discussion on this interaction and MASLD.
Since the major area of discussion is the lipid accumulation in liver this should be well summarized to readers. Figure 2 needs a microscope to be understood, combining bacterial translocation with bile acid metabolism and Kuppfer cells. I doubt there is any real evidence of activity on cell senescence (the Toll-like receptor rise is correlated, like many other factors, to vit. D deficiency). Same case for autophagy.
Pathogenic fecal bacteria were recently elucidated by Takeuchi et al. (Nature 621, 389, 223) but I do not see the clear link with vitamin D. Possibly stress related mechanisms or the modulaton of Paneth cells appear to be more suitable causative factors.
miRNAs are always an exciting topic but unfortunately here they offer a confusing picture. It would be enough to specify which of them may have a crucial role in MASLD. Overall the inflammatory process is very important in this disease but a less messy overview would help.
Overall this contribution is the product of a lot of wok but it does not help to reader to get a full view of the casue of MASLD without spending lot of time.
Rewriting of the paper trying to follow the normal reasoning by a general reader would certainly help. The scheme of the text with the titles of subchapters also needs some rethinking.
Comments on the Quality of English LanguageEnglish is fair quality. Rereading by an English speaking person would be useful
Author Response
Reviewer 2
Comment 1: The paper by Aggeletopoulou et al provides a quite extensive but somewhat confusing overview on vitamin D and liver disease. A detailed presentation on MASLD epidemiology and pathophysiology can be found today in any textbook and the Authors made no effort in establishing a direct link with Vitamin D intake and epidemiology: more in some parts of the word? Less? They did a relatively good job in detailing the metabolic fate of vitamin D although again this has little to do with the topic in discussion.
Response to comment 1: We thank the reviewer for his comments. As a matter of fact, we only dedicated a very small portion of our review to describing the epidemiology and pathophysiology of the disease, as it was beyond the scope of our work. Moreover, a description of vitamin D intake and epidemiology of MASLD, was also out of the scope of our Review, as we totally focused on uncovering the mechanistic insights into the interplay between vitamin D-VDR and MASLD.
Αρχή φÏŒρμας
Comment 2: The vitamin D receptor is an interesting topic although Figure 1 is difficult to follow. It seems to have analogy with the PPAR system (complexed with the RXR). I do not understand whether the lipid changes occurring in MASLD have anything to do with this. Actually an interaction between VDR and the PPAR gamma has been well described apparently in adipose tissue development. I do not see any discussion on this interaction and MASLD.
Response to comment 2: We have added a detail description of figure 1 in the figure legend in order to be easier to follow, as suggested by the reviewer (lines 151-170).
The interaction between the Vitamin D Receptor (VDR) and Peroxisome Proliferator-Activated Receptor gamma (PPARγ) has indeed been extensively documented, particularly in the context of adipose tissue development. VDR, a nuclear receptor activated by vitamin D, and PPARγ, a nuclear receptor involved in adipocyte differentiation and lipid metabolism, have been shown to exert significant cross-talk in various cellular processes, including adipogenesis. Their interaction plays a crucial role in the regulation of adipocyte differentiation, lipid storage, and insulin sensitivity.
However, in figure 1 there is no mention to this receptor because this figure describes how vitamin D becomes activated through the interplay between VDR and RXR. Vitamin D activation involves a series of molecular events that ultimately lead to the modulation of gene expression and this process is presented in figure 1.
Comment 3: Since the major area of discussion is the lipid accumulation in liver this should be well summarized to readers. Figure 2 needs a microscope to be understood, combining bacterial translocation with bile acid metabolism and Kuppfer cells. I doubt there is any real evidence of activity on cell senescence (the Toll-like receptor rise is correlated, like many other factors, to vit. D deficiency). Same case for autophagy.
Response to comment 3: Figure 2 involves all the information related to the mechanisms involved in MASLD pathophysiology and the signaling pathways that critically contribute to this process. Indeed, there is complicated enough, as reviewer supported. Thus, we have added in the figure legend more information on the evidence presented (lines 261-282). Moreover, we have divided the figure 2 into 6 compartments in order to be better apprehending, as suggested.
Regarding the point of the reviewer for cell senescence; a recent experimental study published in the Clin Res Hepatol Gastroenterol (doi: 10.1016/j.clinre.2019.10.007) demonstrated that active vitamin D could alleviate the development of MASLD by blocking cell senescence through the suppression of the p53-p21 signaling pathway in a rat model, providing a novel nutritional therapeutic insight for the disease (lines 301-304).
Regarding the point of the reviewer for autophagy; there are experimental data which have been described in the section “6.2 Autophagy-Related Mechanisms Mediated by Vitamin D-VDR” which describe the impact of vitamin D on autophagy in the context of MASLD (lines 390-423).
Comment 4: Pathogenic fecal bacteria were recently elucidated by Takeuchi et al. (Nature 621, 389, 223) but I do not see the clear link with vitamin D. Possibly stress related mechanisms or the modulaton of Paneth cells appear to be more suitable causative factors.
Response to comment 4: Takeuchi et al., in their review offer a thorough examination of the association between hosts and microorganisms in the context of insulin resistance, highlighting the influence of microbiota on carbohydrate metabolism. This study by using multimodal techniques investigated the interactions between the gut microbiome and metabolic diseases in humans. However, this study has approached the issue from a completely different perspective and the role of vitamin D was completely out of the viewpoint. The Reviewer suggests an intriguing perspective on the interplay between stress, Paneth cells, and the host-microorganism relationships in insulin resistance. Stress has long been implicated in various physiological processes, including metabolic disorders like insulin resistance. The modulation of Paneth cells, which play a crucial role in maintaining gut homeostasis and influencing the gut microbiota, also seems to be a significant factor in this context. Indeed, by targeting stress-related mechanisms or modulating Paneth cell function, it may be possible to influence the composition and activity of the gut microbiota, thereby improving carbohydrate metabolism and ameliorating insulin resistance. However, I am not sure that this point has something to add in the current review.
Comment 5: miRNAs are always an exciting topic but unfortunately here they offer a confusing picture. It would be enough to specify which of them may have a crucial role in MASLD. Overall the inflammatory process is very important in this disease but a less messy overview would help.
Response to comment 5: We agree with the Reviewer that data on the role of miRNAs are still confusing. The reason is why there is only a hypothesis that supplementation with vitamin D may regulate fibrogenic miRNAs in individuals with MASLD; however, there is not enough evidence to support a significant association between vitamin D or VDR related miRNAs with the disease. However, there is ongoing research on assessing the role of such miRNAs in MASLD (doi:10.1186/s13063-019-3241-7), and because of this we have incorporated this section in our review. Beyond vitamin D, miRNA-regulated pathways have been implicated in the progression of MASLD to MASH, indicating their potential as diagnostic and therapeutic tools in this context; however, our review focuses on the vitamin D role in this condition. In the literature there is one experimental study in rats, which suggests a direct association between vitamin D3 supplementation and improved inflammatory, metabolic, and hepatic effects of high fructose corn syrup-induced fatty liver through the miR-33a and mir-200c. In the revised manuscript we have elaborated on this study to highlight the potential of these miRNAs (lines 557-568).
Comment 6: Overall this contribution is the product of a lot of wok but it does not help to reader to get a full view of the casue of MASLD without spending lot of time. Rewriting of the paper trying to follow the normal reasoning by a general reader would certainly help. The scheme of the text with the titles of subchapters also needs some rethinking.
Response to comment 6: MASLD exhibits a multifaceted etiology, wherein genetic predisposition, metabolic dysfunction, and lifestyle factors intricately interplay with the molecular mechanisms driving the pathology. Our review aimed to present the current understanding of the relationship between vitamin D deficiency and MASLD, highlighting the potential implications for clinical practice and future research directions. Moreover, our review focused on uncovering novel mechanistic insights into the interplay between vitamin D-VDR and MASLD. Therefore, the delineation of the MASLD cause, which is indeed of great interest, is out of scope of the current review. We agree with the Reviewer 2 that our review is extended enough, possibly making the apprehension more difficult; thus, the addition of data regarding the cause of MASLD beyond vitamin D may be redundant.
As far as the second point of reviewer, the reasoning of stratifying the data of our review as presented e.g. separated by factors such as liver homeostasis, autophagy, microbiome, endoplasmic reticulum stress, or genetic polymorphisms is because we tried out to offer insights into the molecular mechanisms underlying the relationship between vitamin D and MASLD, providing a deeper understanding of the biological pathways involved, which was the primary objective of our review.
Comments on the Quality of English Language: English is fair quality. Rereading by an English speaking person would be useful
Response to comment: We have edited the manuscript for correct English language.
Reviewer 3 Report
Comments and Suggestions for Authors
MAFLD is a condition linked to vitamin D deficiency. This study by Aggeletopoulou et al. explores how vitamin D impacts metabolic pathways in MAFLD, including modulating lipid metabolism and storage, lessening hepatic necroinflammation, and improving insulin signaling and resistance. Their findings suggest that vitamin D deficiency disrupts these pathways and may contribute to MAFLD development. While more research is needed, vitamin D supplementation may be a promising, favorable approach in the context of MAFLD.
A few minor concerns may be addressed concisely:
1. Dosage and duration of vitamin D supplementation
2. Can the study benefit from analyzing data stratified by factors like disease severity or underlying metabolic conditions?
3. Long-term kinetics of effects and monitoring over time
4. Are there other potential mechanisms by which vitamin D deficiency might contribute to MAFLD beyond the explored pathways?
Comments on the Quality of English Languagenone
Author Response
Reviewer 3
MAFLD is a condition linked to vitamin D deficiency. This study by Aggeletopoulou et al. explores how vitamin D impacts metabolic pathways in MAFLD, including modulating lipid metabolism and storage, lessening hepatic necroinflammation, and improving insulin signaling and resistance. Their findings suggest that vitamin D deficiency disrupts these pathways and may contribute to MAFLD development. While more research is needed, vitamin D supplementation may be a promising, favorable approach in the context of MAFLD. A few minor concerns may be addressed concisely:
Comment 1: Dosage and duration of vitamin D supplementation
Response to comment 1: A paragraph describing data on vitamin D supplementation duration and optimal dosage has been added (lines 615-637).
Comment 2: Can the study benefit from analyzing data stratified by factors like disease severity or underlying metabolic conditions?
Response to comment 2: Indeed, by analyzing data in this manner, researchers can better understand how vitamin D levels and supplementation impact different subsets of patients within the MASLD population. For example, stratification by disease severity may reveal whether vitamin D status correlates with disease progression or severity of liver damage. Similarly, analyzing data based on underlying metabolic conditions, such as obesity or insulin resistance, may elucidate how these factors interact with vitamin D metabolism and influence MASLD pathogenesis. On the other hand, stratifying by factors such as liver homeostasis, autophagy, microbiome, endoplasmic reticulum stress, or genetic polymorphisms can offer insights into the molecular mechanisms underlying the relationship between vitamin D and MASLD, providing a deeper understanding of the biological pathways involved. Considering that our review article aimed to delineate the underlying mechanisms involved between vitamin D and MASLD, we chose the latter stratification criteria in order to align with the study's research goals and hypotheses.
Comment 3: Long-term kinetics of effects and monitoring over time
Response to comment 3: We have added a comment on the critical importance of the long-term kinetics of vitamin D supplementation effects and vitamin D supplementation monitoring over time (lines 638-646).
Comment 4: Are there other potential mechanisms by which vitamin D deficiency might contribute to MAFLD beyond the explored pathways?
Response to comment 4: Vitamin D deficiency may contribute to MASLD pathogenesis and progression through several mechanisms. The key mechanisms including impaired insulin sensitivity, regulation of inflammatory responses, alterations in lipid metabolism, liver fibrosis, impaired immune regulation, dysbiotic mechanisms etc have been described in detail in the current review. Mechanisms beyond the aforementioned have also been described in this review that may also greatly contribute to disease pathogenesis and progression. Such mechanisms include the presence of genetic polymorphisms, and miRNA-regulated pathways which have been strongly implicated in the progression of MASLD to MASH. As per our current understanding and comprehensive review of the literature, we have diligently integrated all potential mechanisms by which vitamin D deficiency is associated with MASLD pathogenesis and progression.
Round 2
Reviewer 2 Report
Comments and Suggestions for Authors
The Authors have done a relatively good job in trying to organize the text better. I appreciate noting that the paper now is somewhat shorter.
The weak part of the paper are the two Figures. Figure 1 is readable and understandable, even without a 20 line explanation. Just a couple of lines willsuffice.
The worse case is Figure 2, totally unreadable and a with a 40 line explanatory note. I personally would take it away totally, but otherwise let the Authors choose the best part of the Figure and write a2-3 explanatory note
Comments on the Quality of English LanguageJust moderate revision
Author Response
Reviewer 2
The Authors have done a relatively good job in trying to organize the text better. I appreciate noting that the paper now is somewhat shorter.
Comment 1: The weak part of the paper are the two Figures. Figure 1 is readable and understandable, even without a 20 line explanation. Just a couple of lines willsuffice. The worse case is Figure 2, totally unreadable and a with a 40 line explanatory note. I personally would take it away totally, but otherwise let the Authors choose the best part of the Figure and write a 2-3 explanatory note
Response to comment 1: We thank the reviewer for his comments. The extended figure legends were added during the revisions process at the request of the reviewer 1. To resolve this dispute, we decided to separate figure 2 in order to be easier for the reader to understand. As a result, the figure legends have been reduced as well. In the revised manuscript figure 2 describes the potential mechanisms of the vitamin D/VDR axis involved in the development and progression of MASLD in the liver whereas the figure 3 describes the extrahepatic mechanisms. We hope you find this change agreeable.